# Large-Sized Nanocrystalline Ultrathin β-Ga_2_O_3_ Membranes Fabricated by Surface Charge Lithography

**DOI:** 10.3390/nano12040689

**Published:** 2022-02-18

**Authors:** Vladimir Ciobanu, Giacomo Ceccone, Irina Jin, Tudor Braniste, Fei Ye, Francesco Fumagalli, Pascal Colpo, Joydeep Dutta, Jan Linnros, Ion Tiginyanu

**Affiliations:** 1National Center for Materials Study and Testing, Technical University of Moldova, Stefan cel Mare Blvd. 168, 2004 Chisinau, Moldova; irina.plesco@cnstm.utm.md (I.J.); tudor.braniste@cnstm.utm.md (T.B.); 2European Commission, Joint Research Centre (JRC), 21027 Ispra, Italy; giacomo.ceccone@ec.europa.eu (G.C.); francesco-sirio.fumagalli@ec.europa.eu (F.F.); pascal.colpo@ec.europa.eu (P.C.); 3Department of Applied Physics, School of Engineering Sciences, KTH Royal Institute of Technology, Hannes Alfvéns väg 12, 11419 Stockholm, Sweden; feiy@kth.se (F.Y.); joydeep@kth.se (J.D.); linnros@kth.se (J.L.); 4Academy of Sciences of Moldova, Stefan cel Mare Blvd. 1, 2001 Chisinau, Moldova

**Keywords:** β-Ga_2_O_3_, Surface Charge Lithography, ultrathin nanomembranes, phase transformation

## Abstract

Large-sized 2D semiconductor materials have gained significant attention for their fascinating properties in various applications. In this work, we demonstrate the fabrication of nanoperforated ultrathin β-Ga_2_O_3_ membranes of a nanoscale thickness. The technological route includes the fabrication of GaN membranes using the Surface Charge Lithography (SCL) approach and subsequent thermal treatment in air at 900 °C in order to obtain β-Ga_2_O_3_ membranes. The as-grown GaN membranes were discovered to be completely transformed into β-Ga_2_O_3_, with the morphology evolving from a smooth topography to a nanoperforated surface consisting of nanograin structures. The oxidation mechanism of the membrane was investigated under different annealing conditions followed by XPS, AFM, Raman and TEM analyses.

## 1. Introduction

Gallium–oxide is a wide bandgap semiconductor material (4.9 eV at 300 K) attracting the interest of researchers owing to its large Baliga’s figure of merit (BFOM) [1], high breakdown electric field, and high temperature operation etc. These characteristics render it a perfect candidate for applications in high-performance power electronics [2,3], where devices constructed from Ga_2_O_3_ bear the advantage of low power losses during high frequency switching in the GHz regime [4]. Recently, β-Ga_2_O_3_ aeromaterial was demonstrated to exhibit very low reflectivity and high transmissivity in an ultrabroadband electromagnetic spectrum ranging from X-band to several THz, indicating a promising future for applications in communication technologies [5].

Ga_2_O_3_ possesses different crystal structures such as corundum (α), monoclinic (β), defective spinel (γ), orthorhombic (ε) and the δ phase being accepted as a form of the orthorhombic phase [6,7,8]. The monoclinic β-Ga_2_O_3_ is considered the most stable one under normal conditions of temperature and pressure [9] and a majority of field-related studies are conducted on β-Ga_2_O_3_.

Currently, there are many technological approaches developed for growing bulk β-Ga_2_O_3_ such as the Verneuil method, the floating zone method, the Czochralski method, the edge-defined film-fed growth method or the Bridgman method [10,11,12,13,14]. For growing thin films of β-Ga_2_O_3_, special techniques such as molecular beam epitaxy (MBE) [15], low pressure chemical vapor deposition (LPCVD) [16], atomic layer deposition (ALD) [17], halide vapor phase epitaxy (HVPE) [18] and metal–organic vapor phase epitaxy (MOVPE) [19] are usually applied. High-quality thin films are also obtained by applying smart-cut [20], or mechanical exfoliation methods [21]. However, most of these techniques involve multiple harmful chemical reactions and are performed under high growth temperatures. Ding et al. [22] reported on the fabrication of gallium–oxide nanostructure by implying wet etching of epitaxial GaN with subsequent annealing. As a result of 950 °C annealing for 5–15 min, the formation of a β-Ga_2_O_3_ shell on GaN nanowires occurs. Low peak to noise intensity ratio of XRD diffractogram and SAED pattern demonstrate the polycrystalline character of the nanowires [22].

Ga_2_O_3_ thin films prove to be a promising material for solar-blind photodetectors due to its large bandgap and UVC absorption [23], transparent electrodes for optical devices [24,25], gas sensors [26], and high-power Schottky barrier diodes with breakdown field exceeding 1 kV [27]. Moreover, Hwang et al. [28] demonstrated the first high-voltage transistor based on β-Ga_2_O_3_ nanomembranes exfoliated from bulk crystals. Using this method [28], nanomembranes of a thickness in the range of 20 to 100 nm and surface area of only few square-micrometers (µm^2^) can be obtained. Taking into account the peculiarities of the exfoliation process, there remains a need to develop methods for highly reproducible fabrication of large surface area nanomembranes. Gallium–oxide presents a promising material for photonics and nonlinear optics [29,30,31], and for the fabrication of bi-dimensional photonic crystals one may apply different techniques such as Surface Charge Lithography, used previously for designing special 2D structures on GaN ultrathin membranes [32].

In this paper, we demonstrate the fabrication of nanoperforated ultrathin β-Ga_2_O_3_ membranes using a cost-effective technological route consisting of two steps. In the first step, the Surface Charge Lithography (SCL) is applied to fabricate GaN nanomembranes as described previously [33,34,35]. In the second step, the GaN nanomembranes are transformed into crystalline β-Ga_2_O_3_ under conditions of thermal treatment in air at 900 °C. The transformation from hexagonal GaN to β-Ga_2_O_3_ phase is demonstrated by investigation of the samples using TEM, AFM, Raman and XPS.

## 2. Materials and Methods

### 2.1. Fabrication of β-Ga_2_O_3_ Membranes

The GaN nanomembranes were fabricated by the SCL approach using epitaxial layers of GaN (2 µm thick) on sapphire with a free charge carrier density of about 5 × 10^17^ cm^−3^. The photolithography process was performed for defining the desired pattern of the regions to be treated with 0.5 keV Ar^+^ ions in a plasma system at a dose of 10^11^ cm^2^. Ion treatment leads to the generation of point defects in the near-surface region which becomes chemically stable against photoelectrochemical (PEC) etching, due to the negative electrical charges trapped by defects. The photoelectrochemical etching was performed in a 0.1 M KOH solution under continuous UV irradiation from a 350 W Hg lamp.

Ultrathin nanomembranes of β-Ga_2_O_3_ were obtained by annealing of GaN nanomembranes at 900 °C for 1.5 h under ambient conditions.

### 2.2. EM Analysis

The morphology of the membranes was studied by using scanning electron microscopy (SEM, Zeiss Gemini Ultra55 Plus, Oberkochen, Germany) at 10 kV. The high-resolution imaging and crystal structure of the membranes were studied by field-emission transmission electron microscopy (FE-TEM, JEOL JEM-2100F; Akishima, Tokyo, Japan) at 200 kV.

### 2.3. XPS Analysis

The surface chemistry was assessed by X-Ray photoemission spectroscopy (XPS). XPS analysis was carried out by means of an Axis Ultra DLD spectrometer (AXIS ULTRA, DLD Kratos Analytical, Manchester, UK) equipped with both Al monochromatic (hν = 1486.6 eV) and non-monochromatic Mg Kα source (hν = 1253.6 eV). The take-off angle (ToA) with respect to the sample normal was 0° for survey and high-resolution (HR) spectra. Surface charging was compensated using low energy (~4 eV) electrons and adjusted using the charge balance plate on the instrument. Selected samples were also analyzed after ion cleaning (Ar^+^, 2 keV, 5 min). The spectra were calibrated setting hydrocarbon C 1 s component at 285.0 eV.

### 2.4. AFM

The topography of the membranes was investigated using the S.I.S.-ULTRA Objective NanoStation II (Rastersonden- und Sensormeßtechnik GmbH, Herzogenrath, Germany). All measurements were performed in non-contact mode. For determination of surface roughness, the results were processed using the software Gwyddion 2.6.

### 2.5. Raman

The Raman studies in this work were performed in backscattering geometry using a Renishaw InVia Qontor confocal microscope (Renishaw plc, Wotton-under-Edge, UK) equipped with a 532 nm laser excitation source. A 100× microscope objective lens with NA = 0.75 was selected to focus the light on the sample surface. The system calibration was performed on a monocrystalline Si wafer with a main peak measured at 520 cm^−1^. A total of 50 spectra collected at 1 s exposure time and 10% laser power were averaged and baseline subtracted. Cosmic Rays Removal tool was applied to the spectra before analysis.

## 3. Results and Discussion

Figure 1 illustrates the SEM images of the fabricated GaN ultrathin membranes and the morphology evolution during the thermal treatment at different temperatures.

As can be seen from Figure 1, the as grown GaN membrane exhibits a very low roughness, with the RMS measuring around 2.08 nm according to AFM investigation. The sample treated at 500 °C exhibits only slight morphological changes compared with the as-grown membrane; however, a thin oxide layer formation is evidenced by XPS as discussed later. According to Yamada et al. [36], the grain formation of Ga_2_O_3_ starts at the dislocation points in the GaN layers grown on different substrates. Large oxide grains formed at 900 °C can be correlated to the surface mass transport mechanism dominating the selective local oxidation at the surface defects. The GaN oxidation is governed by the following reaction:GaN + O_2_ → GaO_x_ + NO_x_,(1)

The RMS according to AFM measurements (Figure 2) equals 4.09 nm, 27.84 nm and 70.87 nm for samples treated at 500 °C, 700 °C and 900 °C, respectively.

The Raman spectra from Figure 3 show that in the as-grown membranes as well as in membranes treated at 500 °C only the GaN phase can be found, with peaks at positions 557 cm^−1^, 568.2 cm^−1^ and 736.2 cm^−1^ which can be attributed to Raman active modes E1(TO), E2(high), and A1(LO) [37], respectively.

Regarding samples treated at 700 °C, the peak positions at 145.5 cm^−1^, 168.9 cm^−1^, 199.8 cm^−1^, 320.7 cm^−1^, 345.4 cm^−1^, 417.2 cm^−1^, 475.4 cm^−1^, 628.9 cm^−1^, 654.5 cm^−1^, and 766.9 cm^−1^ can be attributed to Raman active modes of Ga_2_O_3_ Bg(2), Ag(2), Ag(3), Ag(4), Ag(5), Ag(6), Ag(7), Ag(8), Bg(5) and Ag(10) [38], respectively. The peaks at 568.2 and 736.2 cm^−1^ can be related to the GaN nucleation layer under the membrane. The nucleation layer with a thickness of about 50 nm was grown initially on the sapphire substrate in order to produce a high quality GaN with a thickness of 2 µm. This nucleation layer along with the membrane proved to be chemically stable during the photoelectrochemical etching process. However, in samples treated at 900 °C, only the β-Ga_2_O_3_ phase can be detected, suggesting that even the nucleation layer was completely oxidized in this case.

In Figure 4, the XPS survey spectra of the initial GaN wafer, the as grown GaN ultrathin membrane and the samples annealed at different conditions are presented. It is to be noted that the survey spectra were collected using an Mg source in order to easily quantify the Ga and N species. A slight Pb amount (<0.5 at%) was detected and attributed to the contamination from silver paste used in the PEC process for contacting electrically the GaN surface with the electrode (Figure 4a). A short etching process with Ar was necessary in order to reduce the hydrocarbon contamination below 2 at%. In this case, a decrease of N content could be expected, since Ar ions will preferentially etch N and O resulting in a possible enrichment of the Ga species [39,40]. However, in our experiments, given the short etching time and the low energy Ar beam, we did not observe such an effect. The O 1s photoemission peak in the GaN wafer spectra can be attributed to a thin native oxide layer, whilst the O 1s peak of the as grown film was likely due to the presence of oxygen in the film structure as indicated by the shift to a lower binding energy (~0.5 eV). After annealing, the O 1s peak intensity increases whereas that of the N 1s peak reduces almost to zero (Figure 4a,d). This finding together with the Ga 2p peak shift of about 1 eV toward higher binding energies (Figure 4b) proves the film oxidation. Moreover, the analysis of Ga 3d and Ga 3p (not shown) doublets further corroborate the full oxidation of the film. In particular, the Ga 3d doublet shows a shift of about 1.3 eV toward higher binding energies (Figure 4c), whilst the position of the Ga 3p reported in Table 1 reveals a systematic shift to higher energies as expected in case of Ga oxidation [41,42,43,44].

The as-grown GaN membrane shares a wurtzite type P6_3_mc structure oriented along [1] zone axis, single crystal texture as indicated by electron diffraction (ED) as well as a simulated diffraction pattern (Figure 5a,b). High resolution (HR) micrographs of the membranes treated at three temperatures (Figure 5c,f,h) show a well textured surface; however, with an increase in temperature a partial evaporation and material consumption necessary for recrystallization occurs. After 500 °C treatment (Figure 5d,e), ED indicates the formation of an α-Ga_2_O_3_ R-3c structure oriented along the [1] axis. Additional reflections can be attributed to [1] wurtzite GaN and [111] γ-Ga_2_O_3_ Fd-3m, which coincide in the given zone axis orientations for first and third order of reflections. As XPS results demonstrate the presence of both nitrogen and oxygen species, one can conclude the formation of oxide occurs simultaneously with retention of GaN. ED patterns of 700 °C treated samples (Figure 5g) reveal a clearer picture and are indicative of a higher quality crystal structure. At the same time, the pattern can be attributed to same structures and orientations as at 500 °C treatment. Despite the low temperature treatment, these samples contain an increased level of oxygen and lack of nitrogen as depicted in Figure 4b,c. Therefore, the GaN phase can be excluded and only the formation of α and γ-Ga_2_O_3_ crystal structures is confirmed in 700 °C treated membranes. Analysis of the crystal structure of 900 °C treated membranes (Figure 5h,i) is highly complicated by the recrystallization and formation of disordered nanograins. Polycrystalline diffraction patterns and fast Fourier transformations of HR can be attributed to all GaN, α, β and γ modifications of Ga_2_O_3_. High treatment temperature and XPS results exclude the possibility of preservation of GaN and α-Ga_2_O_3_ modifications. We managed to identify the highly strained [183] oriented β-Ga_2_O_3_ ED pattern presented in Figure 5i.

The crystalline quality of the obtained Ga_2_O_3_ nanomembranes is comparable to that inherent to ultrathin layers of Ga_2_O_3_ obtained by using ALD techniques with subsequent annealing. A recent study reported on ICPE-ALD synthesis of 2D layers of β-Ga_2_O_3_ on Si, sapphire, and glass [45]. The authors demonstrated evolution of XRD patterns from an amorphous layer to a fairly-crystallized one after ex situ treatment at 800 °C. Applying various growth and crystallization methods, it remains difficult to combine large-area uniformity and high crystal quality of β-Ga_2_O_3_ structures. The nanometric thickness of the membranes and the possibility to keep them free-standing, and exclusion of the substrate impact during the growth process represent the key points differentiating this work from other studies of ultrathin layers of Ga_2_O_3_.

## 4. Conclusions

The results obtained in the present study demonstrate that thermal treatment of GaN ultrathin membranes fabricated using SCL leads to a phase transformation to Ga_2_O_3_. The most stable Ga_2_O_3_ phase was obtained by annealing the GaN membranes at 900 °C in air. At intermediate temperatures, the formation of α and γ phases of Ga_2_O_3_ were disclosed by TEM analysis. The AFM demonstrates the formation of nanograins with an increasing size after annealing the samples at temperatures reaching up to 900 °C. Thus, the combination of the SCL and thermal treatment under ambient conditions presents a cost-efficient approach for obtaining large-sized ultrathin membranes of β-Ga_2_O_3_.

## Figures and Tables

**Figure 1 nanomaterials-12-00689-f001:**
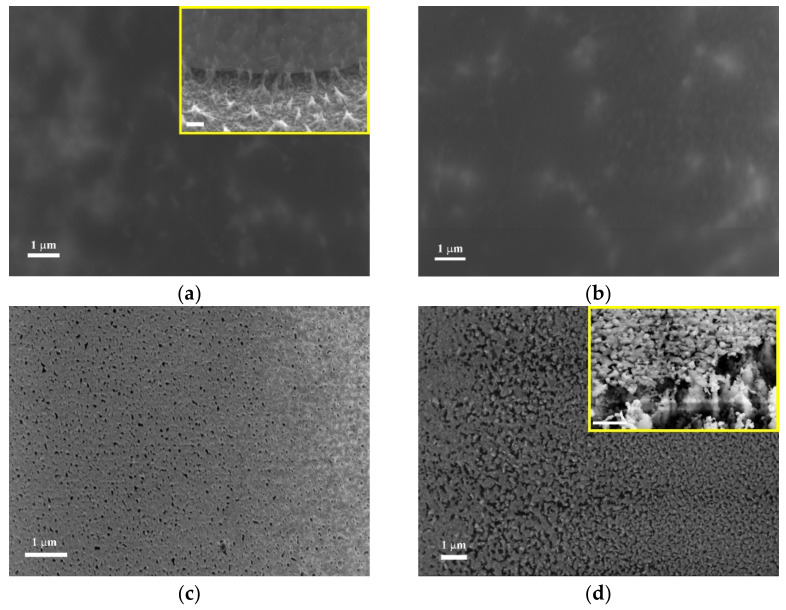
SEM images of the fabricated membranes: (**a**) as-grown GaN membrane and treated at (**b**) 500 °C, (**c**) 700 °C and (**d**) 900 °C. The inserts in (**a**) and (**d**) represent oblique views; the scalebar is 1 μm.

**Figure 2 nanomaterials-12-00689-f002:**
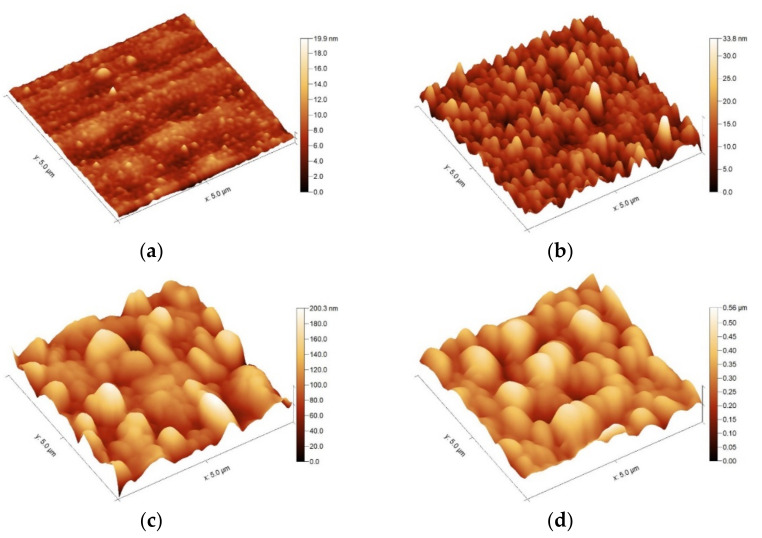
The topography of the as grown (**a**) and thermally treated samples at (**b**) 500 °C, (**c**) 700 °C and (**d**) 900 °C.

**Figure 3 nanomaterials-12-00689-f003:**
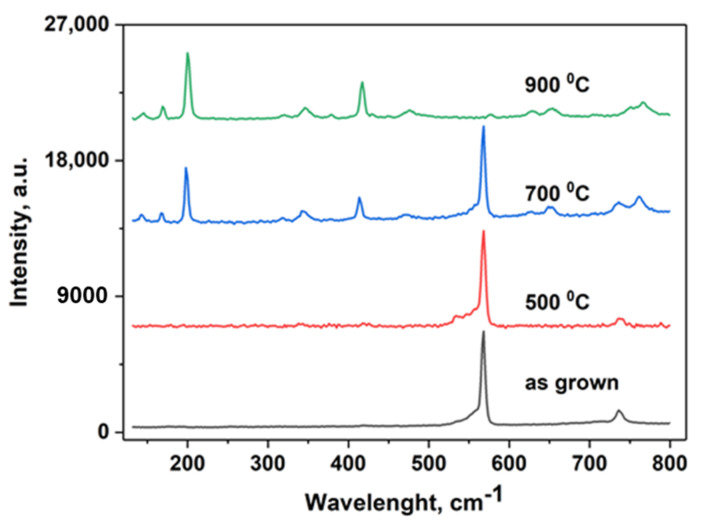
The Raman spectra of as-grown GaN membrane and membranes thermally treated at different temperatures.

**Figure 4 nanomaterials-12-00689-f004:**
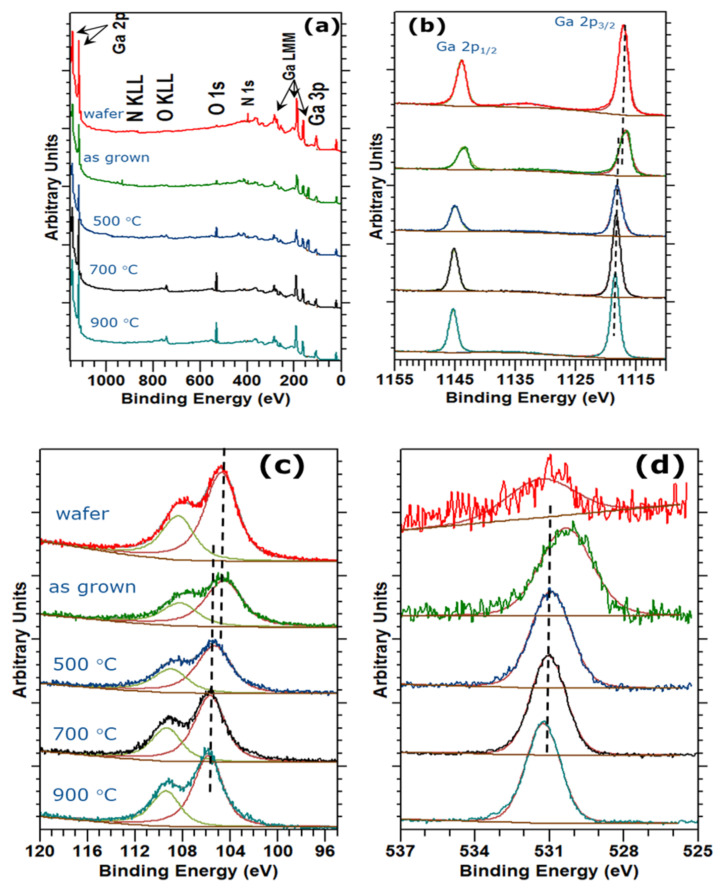
XPS Spectra of the as-grown and thermally treated membranes: (**a**) survey spectra, (**b**–**d**) core level spectra of Ga 2p, Ga 3p and O 1s, respectively. (Samples etched for 2 min with 2 keV Ar+ ions).

**Figure 5 nanomaterials-12-00689-f005:**
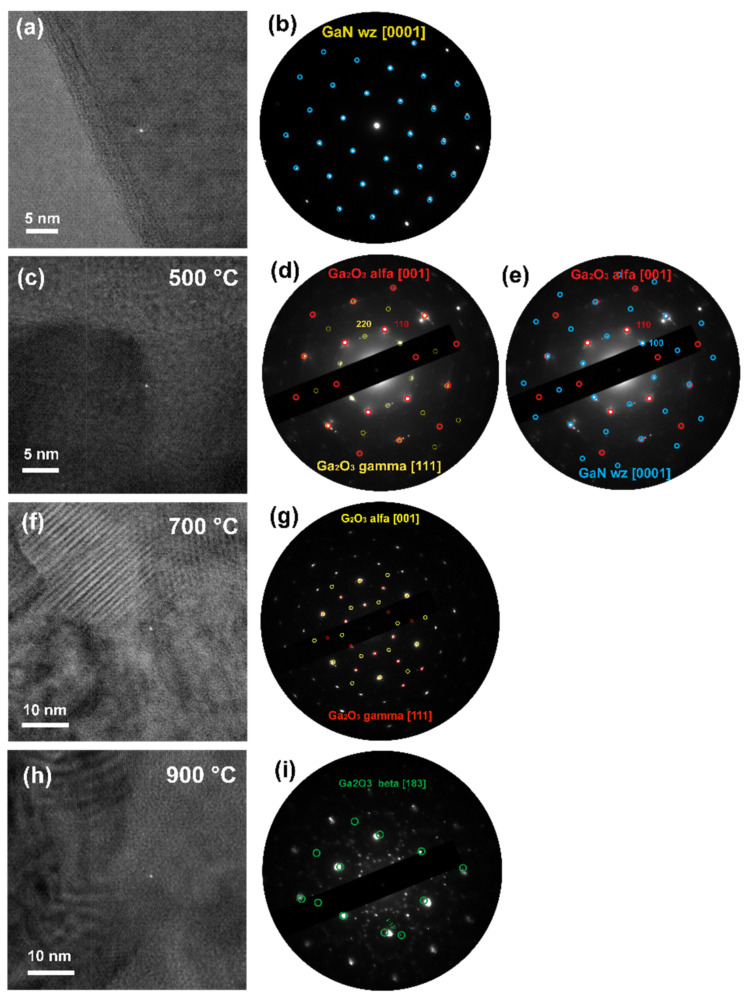
TEM analysis of as-grown and thermally treated membranes: (**a**) HR micrograph of as-grown, (**c**) 500 °C, (f) 700 °C and (**h**) 900 °C treated membranes; (**b**,**d**,**e**,**g**,**i**) ED patterns composed of simulated patterns.

**Table 1 nanomaterials-12-00689-t001:** Ga 3d peak positions obtained by fitting XPS core level spectra.

Sample	Ga 3d Peak Position (eV)
Wafer	19.46
As grown	19.26
500 °C	19.97
700 °C	20.28
900 °C	20.45

## Data Availability

The data presented in this study are available on request from the corresponding authors.

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
