# Peer review of "Large-Sized Nanocrystalline Ultrathin β-Ga2O3 Membranes Fabricated by Surface Charge Lithography"

_nanomaterials, 2022, doi:10.3390/nano12040689_

Round 1

Reviewer 1 Report

The paper under consideration has the potential to be published in the Nanomaterials journal. I have the following suggestions to improve the quality of the paper.

  • In the Introduction section, the author has not explained the potential application of Beta-Ga2O3 membrane? In line 48, it is mentioned that first high voltage transistor has been developed. But what is the authors’ intention behind the fabrication of this layer? Please state the application's point of view.
  • I suggest the author add more literature review on other approaches to obtain Ga2O3 films. And enrich the manuscript with recent relevant papers and compare the fabrication performance with the one proposed in this paper.
  • Why was the annealing process stopped at 900 C? What effect could be seen on the phase transformation? Moreover, what’s the influence of annealing time on the surface roughness and phase?
  • What is the reproducibility of the fabrication process? How many samples were fabricated for each temperature? What is the average surface roughness? Please make a table and mention all these details with respect to different samples.

Reviewer 2 Report

The authors demonstrated the fabrication of nanoscale ultrathin β-Ga2O3 membranes with nanoscale thickness by surface charge lithography. The experiments are well-prepared and the structure of the paper is good. But the introduction part only mentioned MBE, MOVPE to create Ga2O3 thin films. I recommend the publication of this paper after the authors include other methods to create Ga2O3 nanoscale thin films in the introduction part, such as LPCVD, ALD, smart-cut. (APL Materials 7, 022514 (2019) ;  Appl. Phys. Lett. 116, 062105 (2020); ACS Appl. Mater. Interfaces 2020, 12, 40, 44943–44951 )

Additionally, if possible, the authors may compare the quality of their films with the ones in the literature. Here, quality, I mean crystalline or XRD data. 

Round 2

Reviewer 1 Report

I am willing to accept the paper in its current form.